# Effects of Propylene Glycol on Negative Energy Balance of Postpartum Dairy Cows

**DOI:** 10.3390/ani10091526

**Published:** 2020-08-28

**Authors:** Fan Zhang, Xuemei Nan, Hui Wang, Yiguang Zhao, Yuming Guo, Benhai Xiong

**Affiliations:** 1State Key Laboratory of Animal Nutrition, Institute of Animal Science, Chinese Academy of Agricultural Sciences, Beijing 100193, China; 18813015831@139.com (F.Z.); xuemeinan@126.com (X.N.); wanghui_lunwen@163.com (H.W.); zhaoyiguang@caas.cn (Y.Z.); 2State Key Laboratory of Animal Nutrition, College of Animal Science and Technology, China Agricultural University, Beijing 100193, China; guoyum@cau.edu.cn

**Keywords:** propylene glycol, dairy cows, postpartum, negative energy balance, ketosis

## Abstract

**Simple Summary:**

After calving, the milk production of dairy cows increases rapidly, but the nutrient intake cannot meet the demand for milk production, forming a negative energy balance. Dairy cows in a negative energy balance have an increased risk of developing clinical or subclinical ketosis. The ketosis in dairy cows has a negative impact on milk production, dry matter intake, health, immunity, and reproductive performance. Propylene glycol can be used as an important gluconeogenesis in ruminants and can effectively inhibit the formation of ketones. Supplementary propylene glycol to dairy cows during perinatal is an effective method to alleviate the negative energy balance. This review summarizes the reasons and consequences of negative energy balance as well as the mechanism and effects of propylene glycol in inhibiting a negative energy balance in dairy cows. In addition, the feeding levels and methods of using propylene glycol to alleviate negative energy balance are also discussed.

**Abstract:**

With the improvement in the intense genetic selection of dairy cows, advanced management strategies, and improved feed quality and disease control, milk production level has been greatly improved. However, the negative energy balance (NEB) is increasingly serious at the postpartum stage because the intake of nutrients cannot meet the demand of quickly improved milk production. The NEB leads to a large amount of body fat mobilization and consequently the elevated production of ketones, which causes metabolic diseases such as ketosis and fatty liver. The high milk production of dairy cows in early lactation aggravates NEB. The metabolic diseases lead to metabolic disorders, a decrease in reproductive performance, and lactation performance decline, seriously affecting the health and production of cows. Propylene glycol (PG) can alleviate NEB through gluconeogenesis and inhibit the synthesis of ketone bodies. In addition, PG improves milk yield, reproduction, and immune performance by improving plasma glucose and liver function in ketosis cows, and reduces milk fat percentage. However, a large dose of PG (above 500 g/d) has toxic and side effects in cows. The feeding method used was an oral drench. The combination of PG with some other additives can improve the effects in preventing ketosis. Overall, the present review summarizes the recent research progress in the impacts of NEB in dairy cows and the properties of PG in alleviating NEB and reducing the risk of ketosis.

## 1. Introduction

The transition period from late pregnancy to early lactation is well known as the critical time for the production of cows [1]. During this period, the higher energy demand for milk production coupled with the reduction in dry matter intake (DMI) around calving means that a large number of dairy cows are in a state of negative energy balance (NEB). To support the energy requirement, the body fat and protein of dairy cows are mobilized for hepatic gluconeogenesis, which leads to the increase of non-esterified fatty acids (NEFA), β-hydroxybutyrate (BHBA), and ammonia in plasma [2]. Invariably, the dairy cows with high-producing ability have the risk of subclinical ketosis (SCK, hyperketonemia without clinical signs) or clinical ketosis (CK, hyperketonemia with clinical signs). Cows with ketosis have a greater risk of several diseases including displaced abomasum, infections of the reproductive tract, mastitis, cystic ovarian disease, leg problems, and diseases of the digit and foot [3]. Due to the rapid increase in energy demand for milk after calving, the NEB usually has an adverse impact on health and thus decreases animal welfare, production, and profitability [4].

Earlier treatment of ketosis is important to reduce future economic losses in modern high-yield dairy farms. The goal of ketosis treatment is to stimulate gluconeogenesis, increase plasma glucose, and decrease lipolysis [5]. Propionate is the major precursor for gluconeogenesis. However, the limited feed intake during early lactation restricts ruminal propionate supply to the liver, raising the requirement for alternative gluconeogenic precursors [6]. Propylene glycol (PG) is a precursor of ruminal propionate that can be rapidly absorbed from the rumen for gluconeogenesis in the liver [7]. It has long been used as a treatment against ketosis [8]. Experimental studies have shown that an oral drench of PG can be effective in increasing glucose and decreasing BHBA and NEFA in plasma [9]. Therefore, this paper reviews the effects of NEB in dairy cows, and the research progress about the properties of PG in alleviating NEB and reducing the risk of ketosis during postpartum in dairy cows.

## 2. The Formation of NEB in Dairy Cows

During the transition period, dairy cows experience a dramatically increased demand for nutrients for the growing fetus and the initiation of lactation [9]. In the postpartum period, the nutrient requirements for milk yield, milk fat, milk protein, and milk lactose increase rapidly and exceed the supplies from DMI [10]. In addition, the diet of dairy cows after calving changes sharply from being mainly forage-based to concentrate-rich [11]. The sudden energy requirement for milk production and the DMI lags behind, inducing the negative nutrient balance, especially the NEB. The NEB symptoms appear in postpartum, but the dynamic changes of the physiological and metabolic status are verified from the prepartum period [12]. Responding to the NEB, the cow physiologically triggers the lipomobilization of body fat reserves, thereby amounts of NEFA are released into the blood circulation [12]. The NEFA are used as a fuel source by muscle, incorporated into milk fat, and taken up by the liver in proportion to their supply [13]. However, the excessive mobilization of body fat reserves leads to the accumulation of triglycerides in the liver or conversion to ketone bodies (i.e., BHBA, acetone, and acetoacetate) and leads to the onset of ketosis, which has adverse effects on the health, milk productivity, and reproduction in dairy cows. It is well known that dairy cows already go into a period of NEB a few days before calving and that the NEB extends to a few weeks after calving, with the nadir of NEB during the first week of lactation [14]. The feed intake increases slowly at the beginning of lactation [15] and the NEB switches to a positive range after approximately 45 d of lactation [16]. Other diseases such as rumen distention, abomasum displacement, metritis, mastitis, and so on also lead to insufficient nutrient supply and trigger secondary ketosis in the early lactation period. Therefore, methods of decreasing the release of NEFA from adipose tissue are important to alleviate the NEB of dairy cows in early lactation.

## 3. The Effects of NEB in Dairy Cows

### 3.1. Increasing the Incidence of Metabolic Diseases

During the transition period, in order to meet the energy requirement for improved milk production, the rate of lipolysis excesses the lipogenesis, which results in the increasing serum NEFA of dairy cows. When the NEFA could not be completely oxidized to carbon dioxide, it will partly be oxidized to ketone bodies or be stored in the liver as triglycerides [9]. During the period of high metabolic demands, the increased hepatic oxidation of NEFA induces satiety, decreases feed intake, and increases fat mobilization [17]. The blood BHBA level of above 1.2 mmol/L or 1.4 mmol/L is related to impaired health and performance, and is a common and costly metabolic disease, which is called hyperketonemia [18]. The high concentrations of NEFA and BHBA have negative effects on the health and productivity of dairy cows. Therefore, the physiological conditions associated with insufficient energy supply predispose dairy cows to metabolic and microbial diseases such as milk fever, endometritis, ketosis, displaced abomasum, and retained placenta [11].

The epidemiology of ketosis in dairy cows in early lactation increases the risk of displaced abomasum. Each 0.1 mmol/L increases in BHBA at the first SCK-positive test increased the risk of developing a displaced abomasum by a factor of 1.1 [95% confidence interval (CI) = 1.0 to 1.2] and increased the risk of removal from the herd by a factor of 1.4 (95% CI = 1.1 to 1.8) [19]. Raboisson et al. [20] summarized the association between SCK and displaced abomasum in 38 models from 10 publications and found that the risk (95% CI) of left displaced abomasum in cows with SCK were 3.55 (2.60–4.25). Fatty liver is also a metabolic disorder of dairy cows relating to NEB in early lactation. Fatty liver develops when the hepatic uptake of lipids exceeds the oxidation and secretion of lipids by the liver and thereby causes accumulation of triacylglycerol (TAG) in the liver [21]. The severe fatty liver causes metabolic dysfunction, which will reduce the hepatic metabolism, defense function, and insulin sensitivity [12]. The results of Fiore et al. [22] showed that fatty liver already developed before parturition, and increased from moderate to severe in 10 days after calving and then progressively disappeared. Therefore, methods of preventing hepatic lipidosis should be applied during this period.

### 3.2. Decreasing the Milk Productivity Performance of Dairy Cows

McArt et al. [19] concluded that each 0.1 mmol/L increase in BHBA at the first SCK-positive test was associated with a decrease in milk production by 0.5 kg/d for the first 30 days in milk (DIM). The cows with CK were lower in milk production and milk protein content, but milk fat content was higher than healthy cows [23]. A high percentage of fat and a low percentage of protein in the milk were associated with significant increases in the risk of SCK [24]. The mean fat to protein percentage ratio (FPR) and the frequency of FPR > 1.5 were higher in ketosis cows than healthy cows [25]. Therefore, the FPR of milk in early lactation is negatively correlated with energy balance and has been used as an indication of ketosis. The optimal FPR values are 1.05 to 1.18, while FPR values higher than 1.3 or 1.5 suggest a severe NEB and SCK [26]. In the cows with ketosis, the plasma NEFA from fat mobilization provided the precursor for milk fat synthesis in the mammary gland. The results [23] of in vivo and in vitro data indicated that NEFA could induce cell death-inducing DNA fragmentation factor-α-like effector A (CIDEA) expression in bovine mammary epithelial cells, leading to upregulation of de novo fatty acid synthesis enzymes (fatty acid synthase and acetyl-CoA carboxylase 1) and milk lipid secretion proteins (butyrophilin and xanthine dehydrogenase), thereby contributing to an increase in milk fat content in CK cows. The decrease in milk protein percentage might be related to the increased amino acid requirements for gluconeogenesis in ketosis cows, and the spared would be limited for protein synthesis in mammary gland.

### 3.3. Decreasing the Reproductive Performance

The reproductive performance of cows is one of the most important factors affecting the economic benefits of dairy production. The duration and severity of dairy cows’ NEB in early postpartum are also related to reproductive performance. Extensive mobilization of fat has detrimental effects on liver function due to the accumulation of TAG, impairing the detoxification of ammonia into urea [27]. The NEB of dairy cows in early lactation will also increase the mobilization of protein, which will increase the metabolic residues of ammonia and urea. Ammonia is believed to play a role in starting before ovulation, whereas urea mainly interferes negatively after fertilization [28]. The first ovulation of a dairy cow is retarded by decreasing the luteinizing hormone (LH) pulsatility because of the low blood glucose [29].

The calving-to-first-service interval was 8 d longer and the calving-to-conception interval was 16 to 22 d longer in cows with SCK than in healthy cows [20]. Rutherford et al. [30] established that the SCK cows prolonged calving to the first estrus, calving to first insemination and calving to pregnancy intervals, and the first insemination was 4.3 times less likely to be successful compared to non-SCK cows. The high BHBA values, before, after, or before and after artificial insemination were reported associated with a six to 14% reduction in the pregnancy per artificial insemination compared with cows with low BHBA values [31]. Najm et al. [32] also showed the activity of healthy cows exceeded the ketosis cows by an average of 52.6% in 4–70 DIM and the mean motion activity on the day of estrus was also higher in healthy cows. The activity level of the cow will also affect the effective monitoring of estrus, which may be a factor decreasing the reproductive performance, especially detecting estrus by automated surveillance systems. The uterine inflammation could also be exacerbated by the elevated circulating concentration of BHBA or NEFA in early postpartum [33], which would delay uterine involution and successful conception. Therefore, severe NEB will reduce the reproductive performance of cows by delaying uterine recovery, prolonging calving to the first estrus, and reducing estrus activity and successful conception rate.

### 3.4. Inducing Immunosuppression

During the period from late pregnancy to early lactation, the NEB of dairy cows increases the risk of metabolic and infectious diseases. The metabolic status of early-lactating cows is known to affect the immune response to pathogens and impose immune challenges [34]. In this period, the NEB decreases the efficiency in pathogen clearance and increases the magnitude and duration of inflammation [35]. As a consequence, cows are more susceptible to several economically important disease such as metritis and mastitis [36]. In ketosis cows, the inflammation biomarkers of serum amyloid A, haptoglobin, and lipopolysaccharide binding protein are increased when compared with healthy counterparts [37]. The increase in circulating NEFA impairs peripheral blood mononuclear cells and polymorphonuclear leukocytes function, along with a weakening of those cells’ phagocytosis capacity and a decrease in their ability to fight bacteria [38]. The inflammatory state in early lactation may disrupt normal nutrient partitioning and decrease the productivity of dairy cows [39].

Greater concentrations of both NEFA and BHBA have been associated with impaired immune functions and mastitis in dairy cows [40]. Glucose is considered as the preferred substrate for the immune system [41], and the activation of an immune response requires energy [42]. Serum glucose levels in cows with severe NEB are significantly reduced during early lactation, affecting the energy supply of the immune system. The increase in lipid infiltration in the liver also decreases the immune response. Additionally, cows with ketosis (blood BHBA > 3 mmol/L) have higher serum concentrations of proinflammatory cytokines interleukin (IL) 18, tumor necrosis factor (TNF)-α, and IL1B, and lower concentration of anti-inflammatory cytokine IL-10 [43]. The somatic cell count (SCC) in milk is closely related to the immune status of dairy cows. Van Straten et al. [44] concluded that the odds of an event of SCC > 250,000 cells/mL or SCC > 400,000 cells/mL were 44% and 33% greater for cows with ketosis when compared with cows without, respectively. In the results of Abuajamieh et al. [37] ketosis cows had increased circulating markers (serum amyloid A, haptoglobin, and lipopolysaccharide binding protein) of inflammation pre- and post-calving and before the clinical signs of ketosis. The higher prepartum NEFA increases the risk for metritis [45]. The infection and inflammation noticeably redirect resources toward the immune system and away from the utilization and synthesis of economically relevant products [41]. Therefore, severe NEB will lead to fatty liver and high serum NEFA in cows, which also contributes to immunosuppression and increases the risk of infections in the postpartum period. Inflammation postpartum upregulated immune gene expression and mitochondrial uncoupling further increase energy requirements, which exacerbates severe NEB status in cows [11].

## 4. The Anti-Ketogenic Properties of PG and the Mechanism of Inhibiting NEB

During early lactation, glucose synthesis should be increased to accommodate mammary demands [46]. To avoid the occurrence of dairy cow ketosis, it is important to provide extra gluconeogenesis for dairy cows. In 1954, PG was observed to be an effective treatment of ketosis in dairy cows [8]. PG supplementation appears to increase milk yield with a slight decrease in milk fat and an increase in milk lactose percentage [47]. Propylene glycol (1, 2-propanediol; C_3_H_8_O_2_) is a sweet, hygroscopic, viscous liquid that has a gluconeogenic property and is routinely used because of its therapeutic effects on cows suffering from ketosis, based on the premise that it rapidly increases blood glucose [48]. As gluconeogenic precursors, it has been proven that PG is more effective at increasing plasma glucose concentration than glycerol, since 300 mL PG is at least as effective as 600 mL of glycerol [49]. Plasma concentrations of glucose and insulin are known to increase in response to dietary PG [50,51]. Propionate is the main product of PG fermentation, which can be rapidly metabolized with short lag time [52]. This is beneficial for cows to alleviate the NEB and anti-ketogenic. After oral administration, the majority of PG escapes from the rumen wall or gastrointestinal tract and is converted to glucose by the liver [53]. However, the other mechanism of the effects of PG involves the successive production of propionate together with propanal and with the latter being converted to propanol in the rumen, which in turn is converted to propionate and thereafter glucose in the liver [54]. The main effect of PG is to increase the glucogenic status, and as a consequence, the concentration of plasma BHBA is reduced and the cows have decreased risk of developing ketosis [55]. PG is metabolized to lactate, acetate, and pyruvate in the liver. Lactate enters gluconeogenesis via pyruvate, which can be converted to oxaloacetate. The concentration of oxaloacetate is the key metabolite in determining if the acetyl-CoA enters the tricarboxylic acid (TCA)-cycle or ketogenesis [7]. When the oxaloacetate is insufficient for citrate synthase to combine with acetyl-CoA, the excessive acetyl-CoA is then partitioned toward ketone synthesis [37]. The anti-ketogenic properties of PG are partly due to increasing the oxidation of acetyl-CoA into the TCA-cycle and the supply of gluconeogenic glucose [7]. The detailed anti-ketogenic pathways of PG are shown in Figure 1.

In addition, insulin resistance is an adaptation to the very high glucose requirements for lactation, thereby conserving glucose for lactation by limiting its use by insulin-sensitive tissues (muscle, adipose tissue etc.) [56]. The insulin resistance can hence be attributed to a decrease in insulin responsiveness and a decrease in insulin sensitivity [57]. The greater extent of insulin resistance in peripartal dairy cows can contribute to excessive adipose tissue lipolysis and thus greater metabolic disease risk [58]. Chalmeh et al. [59] confirmed that the supplementary feeding with PG reduced the insulin resistance in dairy cows during the transition period by the intravenous glucose tolerance test. The decrease in insulin resistance will inhibit lipolysis and decrease the metabolic disease risk in periparturient dairy cattle. Some researchers have found a negative effect of NEFA in the insulin sensitivity of dairy cattle [57]. Therefore, the effect of PG in decreasing the insulin resistance may be related to the property of PG as a main precursor of glucose and decreasing circulating NEFA.

The energy value of PG is 5.66 Mcal/kg, and according to the assumed PG metabolizable energy utilization efficiency for lactation (80%) of Miyoshi et al. [50], the net energy for lactation (NE_L_) of PG was calculated to be 4.53 Mcal/kg. Due to the higher NE_L_ of PG, it can supply more energy intake than other concentrates for dairy cows in early lactation and reduce the incidence of ketosis.

Therefore, the effects of PG on alleviating NEB in dairy cows are mainly by improving the precursor for hepatic gluconeogenesis and increasing the oxidation of acetyl-CoA into the TCA-cycle. The high energy content of PG can increase the energy density of the diet for dairy cows. The fatty liver and ketone bodies of dairy cows will be inhibited with the increase in liver glucose synthesis.

## 5. Effects of PG on Alleviating NEB in Dairy Cows

### 5.1. The Effects of PG on DMI and Rumen Fermentation

The DMI at the postparturient stage has an essential effect on the NEB metabolism status of cows. The DMI of cows with CK is lower than healthy cows [23]. PG is considered unpalatable and usually reduces feed intake if not mixed thoroughly with other feed components or drenched [7]. Moallem et al. [60] reported the daily average DMI and NE_L_ intake from calving until 100 DIM was higher for cows supplemented with 500 g/d PG per cow until 21 DIM than the control group. The rumen fill score and body condition score (BCS) are also direct tools to evaluate the feed intake and energy balance status. The PG treatment improved the rumen fill score and lowed BCS loss in the dairy cows, which were diagnosed with ketosis in the results of Jeong et al. [61]. The increasing DMI and rumen fill scores are beneficial for decreasing the adverse impact of NEB.

The results of Kristensen and Raun [54] showed that infusion of PG did not affect ruminal pH or the total concentration of ruminal volatile fatty acids (VFA), but decreased the molar proportion of ruminal acetate and increased ruminal concentrations of PG, propanol, and propanal as well as the molar proportion of propionate. Chung et al. [62] also found the PG administration appeared to shift ruminal VFA patterns by producing more glucogenic VFA such as propionate and valerate at the expense of lipogenic VFA such as acetate. The increase of propionate and valerate can provide carbon sources for glucose biosynthesis, which is beneficial for dairy cows to alleviate NEB in early lactation. Acetate is the major source for milk fat synthesis in dairy cows [63]. Therefore, the decrease in acetate concentration may explain the decreased milk fat with PG supplementation.

### 5.2. The Effect of PG on Metabolic Index

It is widely accepted that PG has a glucogenic effect. The glucogenic status of the cows have effects on the liver metabolism of NEFA and, thereby, the regulation of ketogenesis [55]. The effects of PG treatment on SCK or CK have been explained by reduced adipose tissue mobilization, which leads to the decrease of NEFA in the liver and the reduction in the formation of ketone bodies [7]. As Sun et al. [12] reported, the dynamic changes of the physiological and metabolic status were from the prepartum period, so feeding of PG from prepartim is also a good method to alleviate NEB of dairy cows in postpartum. Juchem et al. [64] pointed out that prepartum PG administration increased concentrations of glucose and insulin, and decreased BHBA and NEFA in plasma. It has been validated that the dairy cows suffered deficiency of energy before calving, so the nutritional strategies should be implemented at the start of the prepartum period [12]. The supplement of PG to dairy cows before calving is effective in inhibiting the occurrence of cow ketosis. Therefore, prepartum PG administration has a glucogenic effect for dairy cows in postpartum.

Supplement PG to dairy cows in early lactation is also an important way to avoid energy metabolic diseases. The study of Butler et al. [48] found that drenching 500 mL PG to the dairy cow diet had significant beneficial effect on energy balance and increased plasma insulin and glucose, while the plasma NEFA was decreased. Kristensen and Raun [54] confirmed that when cows were dosed with PG, the plasma concentrations of PG, ethanol, propanol, propanal, glucose, _L_-lactate, propionate, and insulin were increased. Therefore, PG regulates the metabolism of cows by increasing the supply of _L_-lactate and propionate to gluconeogenesis and reducing insulin resistance. Insulin is a key hormone in the regulation of lipolysis in adipocytes. The increase of insulin is also useful for alleviating NEB for dairy cows. Bjerre-Harpøth et al. [55] observed 4-week postpartum PG allocation enhanced glucogenic status, which decreased plasma concentration of BHBA and increased plasma concentration of glucose, but had limited effect on adipose tissue mobilization. Although the metabolic changes in Simmental cows in the periparturient period were not as significant as in the case of Holstein-Friesian cows, the application of PG also resulted in higher milk yield, BCS, and serum glucose content [65]. Therefore, PG can enhance glucogenic status, and decrease the plasma NEFA and BHBA concentrations.

Displaced abomasum, fatty liver, and ketosis are common nutritional metabolic diseases of cows in the postpartum period. PG, as a glucogenic precursor of ruminants, plays an important role in inhibiting metabolic diseases caused by NEB in dairy cows. The results of McArt et al. [66] showed the cows with SCK were 1.6 times more likely to develop displaced abomasum and 2.1 times more likely to be dead or sold than SCK cows treated with PG within the first 30 DIM. The reasons for PG administration decreasing displaced abomasum and the ratio to be removed from the herd are that PG contributes to prevent ketosis and improve milk production. The results of Rukkwamsuk et al. [14] indicated dairy cows drenched with PG from seven days prepartum to seven days postpartum could reduce the risk of fatty liver. This is in accordance with PG decreasing the NEFA in plasma, which will subsequently reduce the TAG accumulation in the liver. Fatty liver is a major metabolic disease of dairy cows in early lactation. The main indicators of hepatic lesions and alterations of its function are the enzymes aspartate transaminase (AST), gamma-glutamyl transferase (GGT), and the blood metabolites glucose, cholesterol, and albumin [67]. The study of Hussein et al. [68] found that PG supplementation has the ability to reduce the enzyme activities of AST and GGT and improve serum glucose, but had no effect on the serum concentrations of total cholesterol and albumin. The PG treatment can thus reduce liver lesions. Stokes and Goff [69] reported offering PG at calving had the effects of lowering the health disorder risk (retained placenta, ketosis, hypocalcemia, displaced abomasum, and metritis etc.) in dairy cows. Feeding PG to SCK cows can effectively prevent the formation of ketone bodies, which will prevent SCK cows developing into CK cows. McArt et al. [70] showed that 300 mL/d of PG treated cows were 1.50 times more likely to resolve their SCK (1.2 ≤ BHBA < 3.0 mM/L) and 0.54 times less likely to develop CK (BHBA ≥ 3.0 mM/L) than the control cows. The reduction in plasma NEFA and increase in plasma glucose are related to the anti-ketogenic property of PG. The use of PG is likely to produce more propionate as the main precursor of glucose; therefore, it can reduce the NEB and insulin resistance [59]. Therefore, the PG supplement to dairy cows can decrease nutritional metabolic diseases in early lactation. The cows drenched with PG could improve the molar proportion of ruminal propionate and hepatic gluconeogenesis, which results in an elevation in serum glucose and a decrease in serum NEFA and BHBA. Drenching of PG during the transition period is therefore beneficial for dairy cows to alleviate NEB in the postpartum period.

### 5.3. The Effects of PG on Milk Production

In the study of Lomander et al. [9], cows supplemented with 300 g of liquid PG daily in the first 21 DIM trended to yield more milk (0.94 kg, 95% CI = −0.03–1.91) compared with control cows during the first 90 DIM, but no difference was found in energy-corrected milk. In the trial of Østergaard et al. [71] based on milk spectra analyses, the results of the treatment with PG (500 mL for 5 d) showed only few benefits in early lactation for physiologically imbalanced cows. The study of Juchem et al. [64] showed that prepartum PG administration had no effect on milk production during the first nine-weeks postpartum. Butler et al. [48] observed that when multiparous Holstein cows received 500 mL oral drench PG from d 10 before expected parturition to d 25 postpartum could increase milk lactose and tended to reduce milk fat content, but there was no difference in milk yield and milk protein percentage. McArt et al. [70] concluded that an oral dose of PG improved milk yield during early lactation in cows with SCK. Stokes and Goff [69] determined that the cows received PG within 4 h of calving and again 24 h post-calving had 3.1 kg/d greater milk production. Therefore, it is conducive to the improvement of postpartum milk performance when cows received PG after calving as soon as possible. PG can provide enough energy to support the increase of milk yield, especially to SCK cows. However, some reports showed there was no difference in milk yield. This may be because the dosages used of PG reduced the feed intake or it was used in the cows without ketosis. The reduced milk fat content affected by PG could be due to the decrease of plasma NEFA and the lower acetate in the rumen [62]. PG had no effect in milk protein possibly because there was no shortage of amino acids for milk protein synthesis. Glucose is necessary for dairy cows to synthesize milk lactose. When PG can supply enough energy and be converted to enough glucose, milk lactose will increase. Therefore, PG tends to increase milk yield and milk lactose, reduce the milk fat of ketosis cows, but has little effect on milk protein.

### 5.4. The Effects of PG on Reproductive Performance

The insufficient energy intake can result in poor reproductive performance such as prolonging postpartum anestrus, decreasing progesterone production by the corpus luteum, and reducing rates of conception [50]. As PG can alleviate the NEB, it therefore also effectively prevents the degradation of reproductive performance.

The results of Gamarra et al. [56] indicated that short-term dietary PG supplementation affected circulating concentrations of metabolites and metabolic hormones, increased progesterone concentrations, and the number of small follicles. The embryo losses were related to the reduced progesterone and the increase in progesterone stimulates and sustains endometrial functions essential for embryonic survival, implantation, and growth [72]. The increase in the number of small follicles is beneficial for early estrus and conception. McArt et al. [66] confirmed that oral administration of PG to SCK cows were 1.3 times more likely to conceive at first insemination than untreated cows. Insulin is necessary for maximal steroidogenesis in both follicular and luteal cells. Miyoshi et al. [50] suggested 500 mL/d of PG administration to NEB dairy cows was able to improve ovarian function in early lactation, which is due to PG induced insulin spike. PG can improve plasma glucose and stimulate insulin secretion. Thus, the increased plasma insulin in PG treated cows has effects on follicular development and LH secretion, leading to earlier ovulation [50]. However, the results of Castañeda-Gutiérrez et al. [73] showed that there was no difference in days to first ovulation in multiparous cows after daily topdressing with PG from last 21 d before expected calving to 21 DIM. Butler et al. [48] observed drenching of PG had no effect on the number of cows with follicles ovulating, undergoing atresia, or becoming cystic, but the day of maximum follicle diameter was earlier for PG treatment. PG treatment advanced the day of maximum follicle diameter, indicating that it promotes follicular growth and is conducive to early estrus.

From the above reports, it can be found that the results for the PG treatment are inconsistent. This may be due to some studies not determining the ketosis status of dairy cows or feeding the PG in diet instead of oral drench. The transient elevations in insulin and glucose, decreases in NEFA, and modest improvement in energy balance are insufficient to adequately stimulate the hypothalamic–pituitary–ovarian axis [48], especially to the dairy cows that do not have a ketosis status. However, to those cows with ketosis, PG can improve plasma glucose and decrease NEFA and therefore effectively improve reproductive performance.

### 5.5. The Effects of PG on Immune Performance

During the peripartum period, dairy cows experience the state of ketosis and fatty liver, which reduces the liver function coupled with increased inflammation and oxidative stress [74]. The cows drenched with PG have a remarkable reduction in TAG accumulation in the liver [14]. There are few direct studies of PG treatment on cow immunity in early lactation. However, metritis and milk SCC can indirectly reflect the immune status of cows. The cows that received PG at calving had significantly lower incidence of metritis [69]. Formigoni et al. [75] observed the mean linear SCC in the first 13 weeks of lactation period was reduced by PG administration (300 g/d from 10 d prior to expected calving until parturition and 300 g/d on days 0, 3, 6, 9, 12 d postpartum). PG is beneficial to increase the serum glucose concentration of postpartum cows and effectively inhibit the risk of fatty liver through gluconeogenesis. Therefore, feeding PG to high-ketone cows can improve the liver function by reducing the accumulation of liver fat, thus improving the immune function of cows.

## 6. The Toxicity and Side Effects of PG

Although PG can prevent ketosis, large doses (>500 g/d) have toxic and side effects on dairy cows due to the toxic compounds of PG during metabolism processes [12]. The clinical signs of PG in toxic doses include depression, ataxia, and excessive salivation as well as abnormal, malodorous, and foul breath and feces in dairy cows [76]. Farmers and veterinarians in Denmark found that some cows expressed the toxicity and side effects of PG [7]. PG toxicity causes oxygen saturation of arterial blood hemoglobin and the oxygen pressure in arterial blood decreases, along with the appearance of dyspnea and ruminal atony upon intake of concentrate containing PG [77]. The sulfur-containing gases produced during PG fermentation in rumen contribute to the toxic effects in rumen when high doses are administered for therapeutic purposes [76]. Hydrogen sulfide is an important signaling substance in hypoxic vasoconstriction, which can explain the link between PG application to the rumen and the dyspnea [77]. The toxicity and side effects of PG limit the maximum dosage in dairy cows to reduce the risk of ketosis. However, the side effects of PG are related to individual cow susceptibility, so it is important to consider the signs of toxicity in the administering of PG, especially at dosages above 500 g per day [7].

## 7. The Feeding Level and Method of PG

The cows drenched with PG could improve the molar proportion of ruminal propionate and hepatic gluconeogenesis, which results in an elevation of serum glucose and decrease in serum NEFA and BHBA. Therefore, the drenching of PG during the transition period is beneficial for the dairy cows to alleviate NEB postpartum. However, the feeding level and method may affect the effects of PG administration.

Gordon et al. [5] pointed out that feeding 300 g of PG daily to ketonic animals should be considered as the base of ketosis treatment. Due to the toxicity and side effects of PG, the maximum feeding level of PG is 500 g per day. The different parity cows have different milk production abilities, so the risk of ketosis is different, which will also affect the PG application strategy. The first lactation heifers had a 47% reduction in the risk of excessive NEB compared to the older cows because of the lower milk yields and lighter bodyweights [78]. Therefore, the use of PG to alleviate NEB should be primarily applied to multiple parity dairy cows.

PG usually exists in liquid form, which is not convenient for feeding in routine by oral drenching to dairy cows. Chung et al. [79] verified feeding PG as a dry product (65% PG and 35% silicon dioxide as the carrier) in total mixed rations (TMR) also reduced plasma BHBA concentration. However, the same amount of PG as a top dress (500 g/d of cookie meal (dried bakery by-product) mixed with dry PG) was more efficient than incorporating it into TMR. The administration method seems to be of importance for the metabolic response of PG in cows because the response of allocating PG as an oral drench or in a separately fed concentrate, is better than mixing into TMR [55]. When PG is added to TMR, the chronic delivery of PG alters the environment in the rumen and inhibits more propionate production, which would decrease the feed intake, increase fat mobilization, and perpetuate the problem of ketosis [5]. Thus, PG is best administered as an oral drench. Meanwhile, feeding PG (mixed with other carriers) in dry product is beneficial to decreasing labor.

Identifying the minimum effective durations for the treatment of ketosis is important for giving PG orally. Gordon et al. [80] observed that extended PG treatment from 3 d to 5 d increased milk production by 3.4 kg/d among the dairy cows that had low blood glucose (<2.2 mmol/L) and 1.7 times more likely to cure with blood BHBA > 2.4 mmol/L, but had no significant effect on milk production in blood glucose ≥ 2.2 mmol/L and cure risk with blood BHBA between 1.2 to 2.4 mmol/L. The additional treatment times to the lower blood BHBA concentration cows might have had little benefit with the stress of increasing labor. McArt et al. [81] confirmed that testing cows 2 days per week from 3 to 9 DIM and treating all positive cows with 300 mL of oral PG for 5 d were the most cost-effective strategy for herds with hyperketonemia incidences between 15% and 50%. For those above 50%, treating all fresh cows with 5 d of PG was the most cost-effective strategy. Therefore, the incidence and degree of hyperketonemia also influence the PG administration durations for treatment. As 75% of cows that developed SCK could be tested as positive within 1 week postpartum (with a peak at 5 DIM) [19], the test of ketones, and treatment of PG should be mainly done in this period. The economic benefit is also a factor influencing the application of PG in preventing ketosis in dairy cows. EI-Kasrawy et al. [82] found that continuous drenching of PG with 300 mL/d for long durations during the transition of dairy cows had a higher net return (1908.52 US$/cow) than drenching for a short duration in 400 mL/d (1171.34 US$/cow) and the control group (1440.21 US$/cow). The economic benefit is affected by the improved production and reproduction performance and the decreased treatment cost of diseases and incidence of early removal from herd. In general, 300 g of PG daily is the basic treatment and the maximum level is 500 g per day. The best feeding method of PG was administered as an oral drench from within one week postpartum and primarily applied to the multiple parities. However, the durations for the treatment of PG for ketosis need to consider the degree of hyperketonemia.

## 8. The Research of Combination Therapy

PG plays an important role in the treatment of ketosis in cows. With 5 d of PG therapy, the rate of cure from hyperketonemia was improved. However, approximately 40% of cows still remained hyperketonemia [66]. The feeding level of PG is restricted due to its potentially toxic effects [8]. Therefore, the use of PG alone still has a poor effect on the treatment of ketosis in some cows. So, the combination of PG with other additives may have a better effect in inhibiting ketosis in dairy cows. There has been a lot of research in this field, but different additives have different effects. The addition of dexamethasone [18] and glycerol [49] to PG showed no additional benefits and the fat appeared to blunt the metabolic response to PG administration of cows [13]. The combination of butaphosphan-cyanocobalamin [80], glucocorticoids [83], L-carnitine-methionine [61], and glucose [84] with PG to the cows of hyperketonemia are beneficial in improving the chances of the resolution of ketosis compared to PG only.

## 9. Conclusions

The NEB in early lactation reduces cow productivity and reproductive performance, and induces immunosuppression, increasing the chance of dairy cows being eliminated. As a precursor of gluconeogenesis, the addition of PG can provide energy and glucose for cows, thus preventing metabolic diseases such as ketosis and fatty liver as well as increasing milk yield and reducing milk fat percentage. PG can also increase the reproductive performance and immune function of cows due to glucose enhancement. However, due to the toxicity and side effects, PG is used in doses within 500 g/d per cow and offered in oral drench. To improve the effects in preventing ketosis, PG is better used with other additives.

## Figures and Tables

**Figure 1 animals-10-01526-f001:**
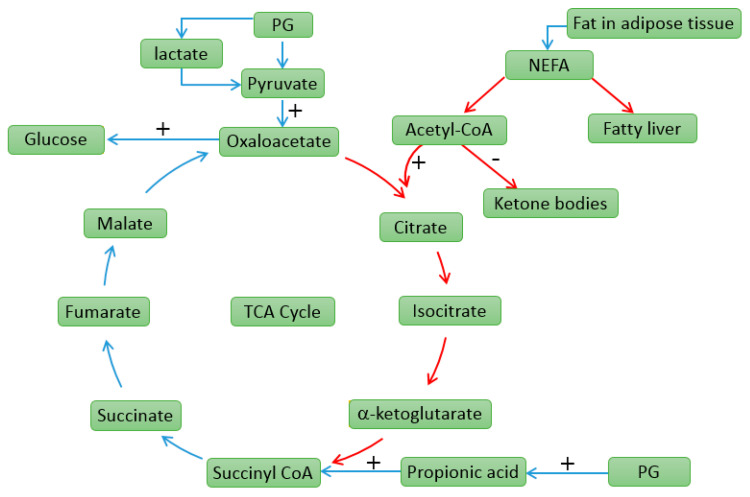
The anti-ketogenic pathways of propylene glycol (PG) in dairy cows [7]. The blue lines are to describe the gluconeogenic pathways of increasing the glucose to preventing ketosis. The red lines are the ways of increasing the oxidation of acetyl-CoA (coenzyme A) in the tricarboxylic acid cycle (TCA-cycle) and the supply of glucose by increasing the production of oxaloacetate, which will prevent acetyl-CoA convert to ketone bodies (β-hydroxybutyrate, acetone, and acetoacetate). PG can also reduce the triacylglycerol (TG) accumulation in liver.

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
