# Peer review of "Effects of Propylene Glycol on Negative Energy Balance of Postpartum Dairy Cows"

_animals, 2020, doi:10.3390/ani10091526_

Round 1
Reviewer 1 Report
The submitted paper is a review regarding the use of propylene glycol in peripertutrient dairy cows. It summarizes the metabolic, health and production periparturient conditions related to negative energy balance and the published research on the effect of propylene glycol on them.
Review summarizes adequately the latest literature, without in depth analysis. In general this theme is well known and much investigated, and still is (regarding periparturient metabolism and related conditions) in the top interest topics of research in dairy cows. However, the present review does not offer anything really new. Even on the most interesting part, that of how to apply the propylene glycol on cows, there are much more publications and is more information available to be offered. By reading this review, the reader can be informed in general about the topic, but on the supplementation/ administration of propylene glycol in cows, more information about its use e.g. in subclinical or clinical cases, its therapeutic dose or dose for prolepsis would be appreciated by the reader.
The big drawback of the review is that English language needs improvement, as in many parts it is not comprehensive.
There are many sentences in that meaning cannot be made out.e.g.L18-19/ L44-45/ L76-77/ L100-101/ L109/ L145-146/ L151-152/ L187-188/ L205-207/ L256/ L259-261/ L315-317/ L323-326/ L382-384/ L402-404
And also many mistakes in language e.g: L17/ L22/L24/ L26/ L31/ L45/ L71-72/ L89/ L121/ L122/ L130 e.t.c.
That makes the text difficult to read and understand
Finally, Figure 1 needs literature reference
Author Response
Response to Reviewer 1 Comments
Thanks so much for your kind comments for this manuscript. We make a detailed modification according to your opinion. The specific modification contents are as follows:
Point 1: Review summarizes adequately the latest literature, without in depth analysis.
Response 1: Some analysis have been added, please see in line 143-145, 205-206, 272-280, 327-330, 339-342, 349-350, 368-399, 406-408, 461-465 et al.
Point 2: The big drawback of the review is that English language needs improvement, as in many parts it is not comprehensive.
There are many sentences in that meaning cannot be made out.e.g.L18-19/ L44-45/ L76-77/ L100-101/ L109/ L145-146/ L151-152/ L187-188/ L205-207/ L256/ L259-261/ L315-317/ L323-326/ L382-384/ L402-404
Response 2:
L18-19: “It is an important method to alleviate negative energy balance of dairy cows in postpartum period by feeding propylene glycol” is changed to “Supplement propylene glycol to dairy cows during perinatal is an important method to alleviate negative energy balance”. Please see in line 18-20.
L44-45: “The NEB of dairy cows increases the body fat and protein are mobilized for hepatic gluconeogenesis to support the energy requirement” is changed to “To support the energy requirement, the body fat and protein of dairy cows are mobilized for hepatic gluconeogenesis”. Please see in line 60-63.
L76-77: “NEFA” is changed to “body fat reserves”. Please see in line 92.
L100-101: “and removal from herd” are deleted. Please see in line 134.
L109: “serve” is changed to “severe”. Please see in line 141.
L145-146: “The peak activity shorter cluster duration in activity clusters associated with first estrus and first insemination postpartum in SCK cows were also lower than non-SCK cows” is deleted, for the result of the study has been mentioned in 191 and 174-188. It is repeat in this place.
L151-152: The uterine inflammation may be exacerbated by the deteriorated inflammation because of NEB” is changed to “The uterine inflammation also may be exacerbated by the elevated circulating concentration of BHBA or NEFA in the early postpartum”. Please see in line195-196.
L187-188: “to alleviate the NEB” are deleted. Please see in line 235.
L205-207: “The anti-ketogenic properties of PG partly be due to increasing the oxidation of acetyl-CoA into TCA-cycle and the increasing the supply of glucose by improving the conversion of oxaloacetate from PG” is changed to “The anti-ketogenic properties of PG partly be due to increasing the oxidation of acetyl-CoA into TCA-cycle and the supply of gluconeogenic glucose”. Please see in line 266-269.
L256: “It is also an important way of feeding PG to postpartum cows to avoid energy metabolic diseases” is changed to “Supplement PG to dairy cows in early lactation is an important way to avoid energy metabolic diseases”. Please see in line 344.
L259-261: “Kristensen and Raun [48] confirmed that the lactating Holstein cows infused with 650 g of PG into the rumen, the plasma concentrations of PG, ethanol, propanol, propanal, glucose, L-lactate, propionate, and insulin increased with PG and the plasma concentrations of acetate and BHBA decreased” is changed to “Kristensen and Raun [54] confirmed that when cows were dosed with PG, the plasma concentrations of PG, ethanol, propanol, propanal, glucose, L-lactate, propionate, and insulin are increased. So the PG affects metabolism of cows by increasing supply of L-lactate and propionate to gluconeogenesis and reducing insulin resistance”, which including the results of the study and analyse. Please see in line347-350.
L315-317: “The NEB of dairy cows in early postpartum period affect the reproductive performance, such as prolong postpartum anestrus, lower progesterone production by the corpus luteum, decrease initial follicular growth and low rates of conception” is changed to “The insufficient energy intake can result in poor reproductive performance, such as prolonging postpartum anestrus, decreasing progesterone production by the corpus luteum, and reducing rates of conception”. Please see in line 437-439.
L323-326: “The increase number of small follicles is beneficial for early estrus and more likely to conceive for cows in early lactation. McArt et al. [63] confirmed the administration of oral PG to SCK cows were 1.3 times more likely to conceive at first insemination than untreated cows, but no difference existed in time to conception within 150 DIM” is changed to “The increase number of small follicles is beneficial for early estrus and conceive. McArt et al. [63] confirmed that oral PG to SCK cows were 1.3 times more likely to conceive at first insemination than untreated cows”. Please see in line 458-460.
L382-384: “The administration of PG is usually as an oral drench or in separately fed concentrate, whereas no metabolic responses also have been observed when mixing PG into TMR” is changed to “The administration method seem to be of importance for metabolic response of PG in cow, for the response of allocating PG as an oral drench or in separately fed concentrate is better than mixing into TMR”. Please see in line 529-532.
L402-404: “The feeding method should be as an oral drench or fed mixed with carriers as dry product separately and primarily applied to the multiple parities from within 1 week postpartum” is changed to “The best feeding method of PG is administered as an oral drench from within 1 week postpartum and primarily applied to the multiple parities”. Please see in line 555-556.
Point 3: And also many mistakes in language e.g: L17/ L22/L24/ L26/ L31/ L45/ L71-72/ L89/ L121/ L122/ L130 e.t.c.
Response 3:
L17: “fertility” is changed to “reproductive performance”. Please see in line 17.
L22: “the feeding levels, methods to alleviating” is changed to “the feeding levels and methods to alleviate”. Please see in line 22.
L24:“increasingly” is changed to “increasing”. Please see in line 26.
L26: The “number” is changed to “amount”. Please see in line 28.
L31: The “improve” is changed to “improves”. Please see in line 33.
L45: “the NEB of dairy cows increases the body fat and protein are mobilized for hepatic gluconeogenesis to support the energy requirement,” is changed to “To support the energy requirement, the body fat and protein of dairy cows are mobilized for hepatic gluconeogenesis”. Please see in line 60-61.
L71-72: “The NEB symptoms are appeared in postpartum” is changed to “The NEB symptoms appear in postpartum”. Please see in line 87.
L89: “During the transition period, the rate of lipolysis excessed the lipogenesis causes the increased NEFA concentration in the blood of dairy cow to support the increasing energy requirement for milk” is changed to “During the transition period, in order to meet the energy requirement for improved milk production, the rate of lipolysis excesses the lipogenesis, which resulted in the increasing serum NEFA of dairy cows”. Please see in line 104-106.
L121: “due to NEB in early lactation” is deleted and the word “precursors” is changed to “precursor”. Please see in line 156.
L122: “indicated” is changed to “indicate”. Please see in line 157.
L130: “The reproductive performance of a cow is one of the most important factors influencing the economic barometers of dairy production” is changed to “The reproductive performance of cows is one of the most important factors affecting the economic benefits of dairy production”. Please see in line 166-167.
L93: “has” is changed to “have” Please see in line 93.
L167: “is” is changed to “are”. Please see in line 167.
L217: “were” is changed to “are”. Please see in line 217.
L227: “increased” is changed to “increase”. Please see in line227.
L250: “properties” is changed to “property”. Please see in line 250.
L255: “was” is changed to “is”. Please see in line 255.
L262: “was” is changed to “is” and “had” is changed to “have”. Please see in line 262.
L292: “had” is changed to “have”. Please see in line 292.
L293: “was” is changed to “is”. Please see in line 293.
L333: “influence” is changed to “effects”. Please see in line 333.
L338: “was” is changed to “is”. Please see in line 338.
L360: “disease” is changed to “diseases”. Please see in line 360.
L365: “decreased” is changed to “decrease”. Please see in line365.
L411: “results” is changed to “result”. Please see in line 411.
L423: “differences” is changed to “difference”. Please see in line 423”.
L533: “ was additive” is changed to “ is added”. Please see in line 533.
L561: “was” is changed to “is”. Please see in line 561.
L600: “effect” is changed to “effects”. Please see in line 600.
Point 4: Finally, Figure 1 needs literature reference
Response 4: There are some content of the fig is same to the reference, so we cited the reference in the title of the Fig. Please see in line 611.
At the same time the fig is also be embellished. Please see in line 612.
All the changes, please see the attachment.

Reviewer 2 Report
The aim of the paper was “to review and systematize the current state of knowledge on effects of propylene glycol in dairy cows”. The authors have collected a large set of studies on the topic, and they describe the results of these studies in a detailed and comprehensive manner. The paper is well-written and a good review of the literature. Some minor comments provided below. In all sections there is not enough summary and synthesis of research (especially chapters: 5.1.; 6; 7; 8).
Specific comments:
- 17. Add immunity.
- 23-24. Not only. This sentence should be rewritten.
- 34. Replace „The feeding method is as an oral drench.”. They administered in the TMR. Consider keeping the sentencje.
- 42-47. Add references.
- 50. What do you mean „add references”?
- 80. What period ? „few days before calving and that the NEB”. Show on the basis of the references.
- 82. Rewrite this sentence. „until around 40 to 80 d of lactation „.There are some aspects that have not been included here that should be:
Grummer, R. R., & Rastani, R. R. (2003). When should lactating dairy cows reach positive energy balance?. The Professional Animal Scientist, 19(3), 197-203.
- 94. Also about the level 1.4 mmol/L.
- 66. Remove „(HK)„.
In chapter 4 indicate the dynamics of changes in blood parameters after administration of PG.
- 187. Indicated the first research:
Johnson, R.B. The treatment of ketosis with glycerol and propylene glycol. Cornell Vet. 1954, 44, 6.
Fisher, L.J., Erfle, J.D., Lodge, G.A. Sauer, F.D. Effects of propylene glycol or glycerol supplementation of the diet of dairy cows on feed intake, milk yield and composition, and incidence of ketosis. Can. J. Anim. Sci.1973, 53, 289-296.
- 213. Replace „spikes”
- 240-242. Sentence not clear. " ... oral drench PG did not affect the concentrations of total VFA, isobutyrate, butyrate and ruminal pH, but the concentrations of propionate, valerate and isovalerate were increased and the concentrations of ammonia, acetate and acetate to propionate ratio were decreased...". Propionic, valeric and isovaleric acids are in group VFA, so concentration of other should be decreased (if total concentration of VFA is not changing);
- 267. Add a discussion of breed differences, e.g.:
Adamski, M.; Kupczyński, R.; Chladek, G.; Falta, D. Influence of propylene glycol and glycerin in Simmental cows in periparturient period on milk yield and metabolic changes. Arch. Anim. Breed. 2011, 53, 238-248.
- 281, Please specify dose of PG.
5.3. The effects of PG on milk production
Have there been studies on the effect of PG on PUFAs in milk ?
5.4. The effects of PG on reproductive performance
Add reference and discussion:
Miyoshi S.; Pate J.L.; Palmquist D.L. Effects of propylene glycol drenching on energy balance, plasma glucose, plasma insulin, ovarian function and conception in dairy cows. Anim Reprod Sci 2001, 68, 29-43.
In L. 354 precise which compounds in PG metabolism are toxic?
- 371 Could you precise what are " kenotic animals"? „ ketonic” ?
- Fig. 1 is a "copy-paste" from
Nielsen, N.I. and Ingvartsen, K.L., 2004. Propylene glycol for dairy cows: A review of the metabolism of propylene glycol and its effects on physiological parameters, feed intake, milk production and risk of ketosis. Animal Feed Science and Technology, 115(3-4), pp.191-213.
Appropriate citation is required.
- The toxicity and side effects of PG
Refine toxic effect.
See chapter 6. Toxicity and side effects of propylene glycol in work: Nielsen, N.I. and Ingvartsen, K.L., 2004. Propylene glycol for dairy cows: A review of the metabolism of propylene glycol and its effects on physiological parameters, feed intake, milk production and risk of ketosis. Animal Feed Science and Technology, 115(3-4), pp.191-213.
Indicate the clinical signs of a toxic dose.
- The research of combination therapy with others
- 405. Remove „with others”
- 414. don't cut it short „HK”
Add comparison PG vs. glycerol. Differences in action.
- 381. Add information in L. 381 "cookie meal (dried bakery byproduct)"
- 417. Remove „severe”.
Cost of preventive use ?
Author Response
Response to Reviewer 2 Comments
Thank you very much for your detailed comments on this article. We have revised and shown each one as follows.
Point 1: 17. Add immunity.
Response 1: The “immunity” is added in line 17.
Point 2: 23-24. Not only. This sentence should be rewritten.
Response 2: “The milk production level has been greatly improved with the continuous improvement of dairy cows feeding level” are changed to “With the improvement in dairy cows intense genetic selection, advanced management strategies, feed quality and improved disease control, the milk production level has been greatly improved”. Please see in line 24-26.
Point 3: 34. Replace „The feeding method is as an oral drench.”. They administered in the TMR. Consider keeping the sentence.
Response 3: In the subsection of “7. The feeding level and method of PG”, I has explained that “When PG was added to TMR, the chronic delivery of PG alters the environment in the rumen and inhibit more propionate production, which would decrease the feed intake, increase fat mobilization and perpetuate the problem of ketosis [5]. Thus, PG is usually best administered as an oral drench”. So the best feeding method is as an oral drench. If mixed the propylene glycol in to TMR, the effect of alleviating NEB will decrease. Please see in line 532-535.
Point 4: 42-47. Add references.
Response 4: The reference of “2. Moore S.M.; DeVries T.J. Effect of diet-induced negative energy balance on the feeding behavior of dairy cows. J Dairy Sci 2020, 103, 7288-7301” is added in line 63 as [2].
Point 5: 50. What do you mean „add references”?
Response 5: This sentence is to express that the disease of ketosis which is induced by negative energy balance has the great risk of other disease, so we should pay close attention to the prevention of ketosis. We have cited the reference of “Lean I.J.: Non-infectious disease: Ketosis. In: Reference module in food science. Elsevier; 2020” in reference 3. Please see in line 66.
Point 6: 80. What period ? „few days before calving and that the NEB”. Show on the basis of the references.
Response 6: This is the basis of the reference,
We can also see the negative energy period in the following fig:
From this fig, we can see that in the a few days before calving, the energy intake is lower than the energy requirement, the dairy cows go into the period of NEB. The nadir of NEB is in the early lactation. Please see in line 94-96.
Point 7: 82. Rewrite this sentence. „until around 40 to 80 d of lactation „.There are some aspects that have not been included here that should be:
Grummer, R. R., & Rastani, R. R. (2003). When should lactating dairy cows reach positive energy balance?. The Professional Animal Scientist, 19(3), 197-203.
Response 7: The sentence “The feed intake increase slowly at the beginning of lactation and until around 40 to 80 d of lactation the NEB switch to a positive range [15]” is changed to “The feed intake increase slowly at the beginning of lactation [15] and until average 45 d of lactation the NEB switch to a positive range [16]”. And the reference of “Grummer, R. R., & Rastani, R. R. (2003). When should lactating dairy cows reach positive energy balance?. The Professional Animal Scientist, 19(3), 197-203” is cited for it summered that “On average, positive energy balance was reached at 45 DIM”. Please see in line 97.
Point 8: 94. Also about the level 1.4 mmol/L.
Response 8: The “concentration” is replaced by “level” and the “greater” is changed to “1.4 mmol/L”. Please see in line 127.
Point 9: 66. Remove „(HK)„.
Response 9: The “(HK)” is removed. Please see in line 129.
Point 10: In chapter 4 indicate the dynamics of changes in blood parameters after administration of PG.
- Indicated the first research:
Johnson, R.B. The treatment of ketosis with glycerol and propylene glycol. Cornell Vet. 1954, 44, 6.
Fisher, L.J., Erfle, J.D., Lodge, G.A. Sauer, F.D. Effects of propylene glycol or glycerol supplementation of the diet of dairy cows on feed intake, milk yield and composition, and incidence of ketosis. Can. J. Anim. Sci.1973, 53, 289-296.
Response 10: The blood parameters change after administration is showed in chapter “5.2 the effect of PG on metabolic index”.
The first research of “Johnson, R.B. The treatment of ketosis with glycerol and propylene glycol. Cornell Vet. 1954, 44, 6” is used in line 237-248 as “In 1954, propylene glycol has been observed as the treatment of ketosis in dairy cows [8]”
The reference of “Fisher, L.J., Erfle, J.D., Lodge, G.A. Sauer, F.D. Effects of propylene glycol or glycerol supplementation of the diet of dairy cows on feed intake, milk yield and composition, and incidence of ketosis. Can. J. Anim. Sci.1973, 53, 289-296.” Is used as “The PG supplementation appeared to increase in milk yield and caused a slight decrease in milk fat and an increase in milk lactose percent [47]” in line 248-249.
The two references are used to express the PG has the property of anti-ketogenic.
Point 11: 213. Replace „spikes”
Response 11: To better explain the effect of PG on insulin resistance, this party is rewrite. “It was concluded that hyperglycemic and hyperinsulinemic effects of PG most likely are caused by insulin resistance induced by increased concentrations of PG and propanol and a decreased ratio of ketogenic to glucogenic metabolites in arterial blood plasma [48]. Chung et al. [51] observed the cows receiving PG had higher spikes in serum insulin concentration during the first 4 h after PG administration. The increasing of insulin will decrease the demand for glucose by peripheral tissues, which in turn reduce lipolysis in adipose tissue and reduces the NEFA from adipocytes and the subsequent ketosis bodies’ production [51].” Is changed to “The insulin resistance can hence be attributed to a decrease in insulin responsiveness and a decrease in insulin sensitivity [57]. The greater extent of insulin resistance in peripartal dairy cows can contribute to excessive adipose tissue lipolysis and thus greater metabolic disease risk [58]. Chalmeh et al. [59] confirmed that the supplementary feeding with PG reduced the insulin resistance in dairy cows during the transition period by intravenous glucose tolerance test. The decrease of insulin resistance will inhibit the lipolysis and decrease the metabolic disease risk in periparturient dairy cattle. Some researchers have found the negative role of NEFA in the insulin sensitivity of dairy cattle [57]. So the effect of PG decreasing the insulin resistance may be related to the property of PG as main precursor of glucose and decreasing circulating NEFA.” Please see in line 273-281.
Point 12: 240-242. Sentence not clear. " ... oral drench PG did not affect the concentrations of total VFA, isobutyrate, butyrate and ruminal pH, but the concentrations of propionate, valerate and isovalerate were increased and the concentrations of ammonia, acetate and acetate to propionate ratio were decreased...". Propionic, valeric and isovaleric acids are in group VFA, so concentration of other should be decreased (if total concentration of VFA is not changing);
Response 12: The Sentence “Chung et al. [51] also found a 200 g /d oral drench PG did not affect the concentrations of total VFA, isobutyrate, butyrate and ruminal pH, but the concentrations of propionate, valerate and isovalerate were increased and the concentrations of ammonia, acetate and acetate to propionate ratio were decreased. Therefore, the PG supplementation can shift the rumen fermentation pattern, and produce more glucogenic VFA (i.e. propionate and valerate)” are simplified to “Chung et al. [62] also found the PG administration appeared to shift ruminal VFA patterns by producing more glucogenic VFA, such as propionate and valerate, at the expense of lipogenic VFA, such as acetate.”
We also add “The increasing of propionate and valerate can provide carbon sources for glucose biosynthesis which are benefit for dairy cows to alleviation NEB in early lactation. Acetate is the major source for milk fat synthesis in dairy cow [63]. So the decreasing of acetate concentration maybe relation to the PG supplementation caused decrease in milk fat.”
We also explain the results of the research. Please see in line 326-331.
Point 13: 267. Add a discussion of breed differences, e.g.:
Adamski, M.; Kupczyński, R.; Chladek, G.; Falta, D. Influence of propylene glycol and glycerin in Simmental cows in periparturient period on milk yield and metabolic changes. Arch. Anim. Breed. 2011, 53, 238-248.
Response 13: The reference is used by added “Although the metabolic changes in Simmental breed cows in the periparturient period are not as significant as in the case of Holstein-Friesian breed, the application of PG also resulted in higher milk yield, BCS and serum glucose content [65]” in line 355-357. This reference major to express the application of PG in Simmental breed cows, it also has the effect of alleviating NEB.
Point 14: 281, Please specify dose of PG.
Response 14: “300 mL/d of” is added in line 404.
Point 15: 5.3. The effects of PG on milk production
Have there been studies on the effect of PG on PUFAs in milk ?
Response 15: we searched the research about the effects of PG on PUFAs in milk on Web of science and PubMed, only found one research: Mesilati-Stahy, R., Malka, H., & Argov-Argaman, N. (2015). Influence of glucogenic dietary supplementation and reproductive state of dairy cows on the composition of lipids in milk. Animal, 9(6), 1008-1015. But the cows allocated to this study were at 60 days in milk. In this stage, cows are transitioning from NEB to positive energy balance, which is not consistent with this manuscript.
Point 16: 5.4. The effects of PG on reproductive performance
Add reference and discussion:
Miyoshi S.; Pate J.L.; Palmquist D.L. Effects of propylene glycol drenching on energy balance, plasma glucose, plasma insulin, ovarian function and conception in dairy cows. Anim Reprod Sci 2001, 68, 29-43.
Response 16: The reference and discussion is added as “Insulin is necessary for the maximal steroidogenesis in both follicular and luteal cells, so Miyoshi et al. [50] suggested 500 ml /d of PG administration to NEB dairy cows increase improves ovarian function in early lactation is due to the PG induced insulin spike. The PG can improve the plasma glucose and stimulate insulin secretion. Thus, the increased plasma insulin in PG treated cows have effects on follicular development and LH secretion, leading to earlier ovulation [50]” in line 461-466.
Point 17: In L. 354 precise which compounds in PG metabolism are toxic?
Response 17: In line 500-502, we mentioned that “The sulfur-containing gases produced during PG fermentation in rumen contribute to the toxic effects in rumen when high doses are administered for the therapeutic purposes” and in line 502-503 we mentioned “Hydrogen sulfide is an important signaling substance in hypoxic vasoconstriction, which can explain the link between PG application to the rumen and the dyspnea”. So the compounds in PG metabolism are toxic is the largely sulphur –containing (H2S) production and lead to the appearance of dyspnea.
Point 18: 371 Could you precise what are " kenotic animals"? „ ketonic” ?
Response 18: the word “kenotic” is replaced to “ketonic”. Please see in line 519.
Point 19: Fig. 1 is a "copy-paste" from
Nielsen, N.I. and Ingvartsen, K.L., 2004. Propylene glycol for dairy cows: A review of the metabolism of propylene glycol and its effects on physiological parameters, feed intake, milk production and risk of ketosis. Animal Feed Science and Technology, 115(3-4), pp.191-213.
Appropriate citation is required.
Response 19:
There are some content of the fig is same to the reference, so we cited the reference in the title of the Fig. Please see in line 611.
At the same time the fig is also be embellished. Please see in line 616.
Point 20: The toxicity and side effects of PG
Refine toxic effect.
See chapter 6. Toxicity and side effects of propylene glycol in work: Nielsen, N.I. and Ingvartsen, K.L., 2004. Propylene glycol for dairy cows: A review of the metabolism of propylene glycol and its effects on physiological parameters, feed intake, milk production and risk of ketosis. Animal Feed Science and Technology, 115(3-4), pp.191-213.
Response 20: the toxic effect of PG in dairy cows are showed in line 389-391. The reference of “Nielsen, N.I. and Ingvartsen, K.L., 2004. Propylene glycol for dairy cows: A review of the metabolism of propylene glycol and its effects on physiological parameters, feed intake, milk production and risk of ketosis. Animal Feed Science and Technology, 115(3-4), pp.191-213” is used by adding “Farmers and veterinarians in Denmark have experienced that some cows expressed the toxicity and side effects of PG [7]” as a case of PG toxicity and side effects. Please see in line 496-497.
Point 21: Indicate the clinical signs of a toxic dose.
Response 21: “It can lead to depression, ataxia, and excessive salivation, as well as abnormal, malodorous, and foul breath and feces” is replaced to “The clinical signs of PG in toxic dose is that it lead to depression, ataxia, and excessive salivation, as well as abnormal, malodorous, and foul breath and feces in dairy cows”. Please see in line 494-496.
Point 22: The research of combination therapy with others
- Remove „with others”
Response 22: The words “with others” were removed. Please see in line 559.
Point 23: 414. don't cut it short „HK”
Add comparison PG vs. glycerol. Differences in action.
Response 23: The word “HK” is all changed to “hyperketonemia”. Please see in line 545, 547, 558, 561,562, 592.
The comparison of PG vs glycerol is added as “As the gluconeogenic precursors, it is proved that PG is more effective at increasing plasma glucose concentration than glycerol and the 300 ml PG is at least as effective as 600 ml of glycerol [47]” in line 252-254. From the research we can see the PG has beet effect than glycerol.
Point 24: 381. Add information in L. 381 "cookie meal (dried bakery byproduct)"
Response 24: “but top-dressing PG (500 g/d of cookie meal mixed with dry PG)” is changed to “but the same amount of PG as a top dress [500 g/d of cookie meal (dried bakery by-product) mixed with dry PG]”. Please see in line 529-530.
Point 25: 417. Remove „severe”.
Cost of preventive use ?
Response 25: The word “serve” is removed. Please see in line 595.
The cost of preventive use is by adding “The economic benefit is also a factor influence the application of PG in preventing ketosis in dairy cows. EI-Kasrawy et al. [82] found the continuous drenching of PG with 300 ml /d for long duration during the transition of dairy cows had the higher net return (1908.52 US$/cow) than drenching for short duration in 400 ml/d (1171.34 US$/cow) and control group (1440.21 US$/cow) . The economic benefit is affected by the improving production and reproduction performance and the decreasing treatment cost of diseases and incidence of early removal from heard.” from the reference “El-Kasrawy N.I.; Swelum A.A.; Abdel-Latif M.A.; Alsenosy A.E.A.; Beder N.A.; Alkahtani S.; Abdel-Daim M.M.; Abd El-Aziz A.H. Efficacy of different drenching regimens of gluconeogenic precursors during transition period on body condition score, production, reproductive performance, subclinical ketosis and economics of dairy cows. Animals (Basel) 2020, 10” as a case of the cost of PG application. Please see line 550-555.
All the changes, please see the attachment.

Reviewer 3 Report
This is an interesting, informative and well detailed study. The objectives of the trial are of interest and fit well within the scope of the Journal. However, in my opinion some aspects should be revised.
The main indicators of hepatic lesions and alterations of its function are the enzymes aspartate transaminase (AST), gamma-glutamyl transferase (GGT) and the blood metabolites glucose, cholesterol and albumin- please describe their changes depending on the use of propylene glycol.
I would introduce the following subsection with the following specifications:
Effects of Propylene Glycol on:
- Body condition score
- First oestrus postpartum
- Milk yield
- Yield of ECM
- Plasma glucose concentration
- Plasma NEFA concentration
- Plasma BHBA concentration
Cases of veterinary-treated or farmer-recorded diseases from 0 to 90 d, where cows were fed Propylene Glycol:
- Ketosis
- Mastitis
- Paresis
Author Response
Response to Reviewer 3 Comments
Thanks very much for your detailed comments on this article. We revised it and shown each one as follows.
Point 1: The main indicators of hepatic lesions and alterations of its function are the enzymes aspartate transaminase (AST), gamma-glutamyl transferase (GGT) and the blood metabolites glucose, cholesterol and albumin- please describe their changes depending on the use of propylene glycol.
Response 1: “ Fatty liver is a major metabolic disease of dairy cows in early lactation. The main indicators of hepatic lesions and alterations of its function are the enzymes aspartate transaminase (AST), gamma-glutamyl transferase (GGT) and the blood metabolites glucose, cholesterol and albumin [67]. The study of Hussein et al. [68] found that the PG supplementation has the ability to reduce enzyme activities of AST and GGT values and improve serum glucose, but had no effect on the serum concentrations of total cholesterol and albumin.. So the PG treatment can reduce liver lesions” are added in line 369-400 to explain the effects of PG treatment on fatty liver.
Point 2: I would introduce the following subsection with the following specifications:
Effects of Propylene Glycol on:
First oestrus postpartum
Milk yield
Yield of ECM
Plasma glucose concentration
Plasma NEFA concentration
Plasma BHBA concentration
Cases of veterinary-treated or farmer-recorded diseases from 0 to 90 d, where cows were fed Propylene Glycol:
Ketosis
Mastitis
Paresis
Response 2: In this manuscript, we introduce the effects of PG on alleviating NEB in dairy cows as the similar indicators are in the same subsection, which are consistent with the effect of NEB in dairy cows in chapter 3.
Such as the indicators body condition score in chapter 5.1 The effects of PG on DMI and rumen variable; ketosis, plasma glucose concentration, plasma NEFA concentration and plasma BHBA concentration in chapter 5.2 The effect of PG on metabolic index; first oestrus postpartum in chapter 5.4 The effects of PG on reproductive performance; the milk yield and ECM in chapter 5.3. The effects of PG on milk production. The PG has the effect of decreasing milk SCC in line 486-490, which showed the PG can prevent mastitis.
All the changes, Please see the attachment.

Reviewer 4 Report
The review by Zhang et al. it is well written and acceptable for publication.
If possible, the authors could extend the argument in section 4.1 and section 8 of metabolic diseases and associated treatments. I suggest articles to read to expand these sections, especially regarding ketosis, lipidosis and metabolic pathologie
Fiore, E.; Piccione, G.; Perillo, L.; Barberio, A.; Manuali, E.; Morgante, M.; Gianesella, M. Hepatic lipidosis in high-yielding dairy cows during the transition period: Haematochemical and histopathological findings.
Anim. Prod. Sci. 2017, 57, 74–80.
Sordillo, L.M.; Contreras, G.A.; Aitken, S.L. Metabolic factors aecting the inflammatory response of periparturient dairy cows. Anim. Health Res. Rev. 2009, 10, 53–63.
Fiore, E.; Perillo, L.; Piccione, G.; Gianesella, M.; Bedin, S.; Armato, L.; Giudice, E.; Morgante, M. Effect of combined acetylmethionine, cyanocobalamin and -lipoic acid on hepatic metabolism in high-yielding dairy cow. JDR 2016, 83, 438–441.
In addition, I suggest removing references older than 20 years from the text and replacing them with more recent articles. Here some examples of old articles in your manuscript:
Herdt et al., 1992 (ref 4)
Bell et al., 1995 (ref 9)
Studer et al., 1993 (ref 45)
Cozzi et al., 1993 (ref 58)
Formigoni et al., 1996 (ref 63)
Author Response
Response to Reviewer 4 Comments
Thank you very much for your valuable comments on this manuscript. According to your comments, we have made detailed modifications to the manuscript as shown below.
Point 1: If possible, the authors could extend the argument in section 4.1 and section 8 of metabolic diseases and associated treatments. I suggest articles to read to expand these sections, especially regarding ketosis, lipidosis and metabolic pathologie
Fiore, E.; Piccione, G.; Perillo, L.; Barberio, A.; Manuali, E.; Morgante, M.; Gianesella, M. Hepatic lipidosis in high-yielding dairy cows during the transition period: Haematochemical and histopathological findings.
Anim. Prod. Sci. 2017, 57, 74–80.
Sordillo, L.M.; Contreras, G.A.; Aitken, S.L. Metabolic factors affecting the inflammatory response of periparturient dairy cows. Anim. Health Res. Rev. 2009, 10, 53–63.
Fiore, E.; Perillo, L.; Piccione, G.; Gianesella, M.; Bedin, S.; Armato, L.; Giudice, E.; Morgante, M. Effect of combined acetylmethionine, cyanocobalamin and a-lipoic acid on hepatic metabolism in high-yielding dairy cow. JDR 2016, 83, 438–441.
Response 1:
(1) The reference of Fiore et al. 2017 is added as ““The results of Fiore et al. [22] showed that fatty liver already develops before parturition, and increase from moderate to serve in 10 days after calving and then progressively disappears. So the methods of preventing hepatic lipidosis should be applied during this period” to demonstrate the change of fatty liver around parturition. Please see in line 143-145.
(2) The reference of “Sordillo, L.M.; Contreras, G.A.; Aitken, S.L. Metabolic factors affecting the inflammatory response of periparturient dairy cows. Anim. Health Res. Rev. 2009, 10, 53–63” is about inflammatory response of periparturient dairy cows. So we used it in line 206-207 to express the relation of NEB to disease as “As a consequence, dairy cattle are more susceptible to several economically important disease such as metritis and mastitis” to explain the inflammatory disease which is relation to NEB. Please see in line 206-207,
(3) The reference of Fiore et al. 2017 is added in section 4 by “During the early lactation, the glucose synthesis should be increased to accommodate mammary demands”. Please see in line 235-236.
In this manuscript, we major reviewed the treatment of PG to NEB. So in section 5, we review the effects of PG on alleviating NE; in section 6, we reviewed the negative effect of large dose use of PG; and in section 7 and 8, we reviewed the feeding methods.
Point 2: In addition, I suggest removing references older than 20 years from the text and replacing them with more recent articles. Here some examples of old articles in your manuscript:
Herdt et al., 1992 (ref 4)
Bell et al., 1995 (ref 9)
Studer et al., 1993 (ref 45)
Cozzi et al., 1993 (ref 58)
Formigoni et al., 1996 (ref 63)
Response 2:
(1) Herdt et al., 1992 (ref 4)
The reference of Herdt et al., 1992 (ref 4) is changed to the reference “Gordon J.L.; LeBlanc S.J.; Duffield T.F. Ketosis treatment in lactating dairy cattle. Veterinary Clinics of North America: Food Animal Practice 2013, 29, 433-445” which has the same content (Providing glucose, stimulating gluconeogenesis, and decreasing fat breakdown form the foundation for rational ketosis treatment). Please see in line 71.
(2)Bell et al., 1995 (ref 9)
The sentence “In the transition period, nutrient requirements of fetus reach maximal levels in the three weeks prepartum, yet DMI decreases by 10-30%,” is changed to “During the transition period, the dairy cow experiences a dramatically increased demand for nutrients for the growing fetus and the initiation of lactation [9]”. And the reference is changed to “Lomander H. et al. 2012” which is published in 2012. Please see in line 81-82.
(3) Studer et al., 1993 (ref 45)
“Studer et al. [48] reported the prepartum PG administration reduced hepatic triglyceride accumulation by 32% and 42% at 1 and 21 d postpartum, respectively.” Is changed to the discussion of PG application in prepartum. “It has been validated that the dairy cows suffered deficiency of energy before calving, so the nutritional strategies should be implemented since the prepartum period [12]. The supplement of PG to dairy cows before calving is good for inhibiting the occurrence of cow ketosis.” The reference is changed to “Sun et al. 2020”. Please see in line 341-342.
(4) Cozzi et al., 1993 (ref 58)
“The study of Cozzi et al. [66] showed the use of PG to dairy cows resulted in a metabolic pattern more favorable to live weight gain than milk production. Weight gain indicates that PG reduces the mobilization of fat from adipose tissue and effectively alleviates NEB of dairy cows” is changed to “The use of PG is likely to produce more propionates as the main precursor of glucose; therefore, it can reduces the NEB and insulin resistance [59]. So the supplement of PG to dairy cow can subsequently decreases nutritional metabolic disease in early lactation” which use the reference of “Chalmeh et al. 2020” and discussion the reason for PG prevent nutritional metabolic disease in dairy cows. Please see in line 407-409.
(4) Formigoni et al., 1996 (ref 63)
For there is little research in the field of PG on immune performance, so the reference is kept in the manuscript.
All the changes, Please see the attachment.

Round 2
Reviewer 1 Report
Αuthors took in consideration all comments and replied accordingly. Significant improvement and changes were incorporated to the text, so it is much more comprehensive now. Text still needs editing from a professional english language editor.
Author Response
Response to Reviewer Comments
Point: Text still needs editing from a professional English language editor. English language and style are fine/minor spell check is required.
Response: Thank you very much for your kind comments. We have revised the English grammar in detail. The details are shown below.
Line:
- 19: “important” is changed to “effective”.
- 20: “reason” is changed to “reasons”.
- 20: “as well as” is added.
- 21: “in” is added”
- 22: “ At the same time” is changed to “in addition”
- 22: “by propylene glycol” is changed to “ using propylene glycol”
- 23: “summarized” is changed “discussed”.
- 24: “cows” is changed to “cows’”.
- 24: “improved” is changed to “are improved”.
- 27: “needs” is changed to “demand”.
- 28: “the production” is changed to “consequently the elevated production”.
- 37: “of” is changed to “in”.
- 38: “on” is changed to “in”.
- 43: “to be” is changed to “as”.
- 44: “demands” is changed to “demand”.
- 62: “the diseases” is changed to “several diseases including”
- 64: “will have” is changed to “usually has”.
- 65: “decrease” is changed to “decreases”.
- 66: “in modern high-yield dairy farms” is added”.
- 69: “limits” is changed to “restricts”
- 71:“Up to now, it” is changed to “It”.
- 73: “Therefore” is added”
- 77: “cow” is changed to “cows”
- 81: “diets” is deleted.
- 87: “apparently” is deleted.
- 88: “converted” is changed to “conversion”.
- 89: “have” is changed to “has”.
- 93: “switch” is changed to “switch”.
- 93: “until average 45 d. is changed to “after approximately 45 d of lactation”.
- 101: “result” is changed to “results”.
- 104: “decreasing” is changed to “decreases”.
- 151: “increasing” is changed to “increases”
- 151: “above” is added.
- 159: “and” is added”.
- 166: “decline” is changed to “reduce”.
- 167: “develops” is changed to “developed”.
- 169: “disappears” is changed to “disappeared”
- 213: “DIM 4-70” is changed to “4-70 DIM”.
- 216: “may” is changed to “could”.
- 219: “and” is added”.
- 222: “In the time” is changed to “During the period”.
- 224: “time” is changed to “period”
- 225: “decrease” is changed to “decreases”.
- 226: “dairy cattle” is changed to “cows”.
- 228: “were” is changed to “are”.
- 238: “decreased” is changed to “decreased”
- 239: “ had “is changed to “have”
- 250: “increased” is changed to “increases”.
- 256: “providing” is changed to “to provide”
- 257: “PG has been observed as the” is changed to “PG was observed as an effective”.
- 280: “appear” is changed to “appears”.
- 285: “and the” is changed to “since”.
- 287: “and it is” is changed to “which can be”.
- 302: “Fig” is changed to “Figure”.
- 311: “role” is changed to “effect”.
- 312: “so” is changed to “Therefore”.
- 318: “major” is changed to “mainly”.
- 324: “have” is changed to “has”.
- 325: “generally is may” is changed to “usually”.
- 329: “tool” is changed to “tools”
- 358: “are benefit” is changed to “is benefical”.
- 358: alleviation” is changed to”alleviate”.
- 360: “caused decrease” is changed to “explain the decreased milk fat with”.
- 379: “good” is changed to “effective in”.
- 374: “also” is added.
- 379: “are” is changed to “were”.
- 386: “breed” is changed to “cows”.
- 395: “help” is changed to “contributes to”.
- 450: “determined” is changed to “reported offering”.
- 456: “relate” is changed to “are related”.
- 462: “therefore” is added.
- 464: “for” is changed to “in”.
- 473: “percent” is changed to “percentage”.
- 478: “But” is changed to “However”.
- 489: “Therefore” is changed to “because”
- 489: “it therefore also” is added.
- 548: “administration of” is added.
- 561: “research result” is changed to “reports”.
- 583: “is that it lead to” is changed to “include”.
- 584: “have experienced that “found”.
- 618: “the treatment of” is changed to “the feeding”.
- 625: “thus it” is changed to “which”.
- 634: “Feeding” is changed to “Meanwhile, feeding”.
- 635: “benefit” is changed to “beneficial”.
- 644: “; and” is changed to “. And for those”
- 653: “improving” is changed to “improved”.
- 653: “decreasing” is changed to “decreased”.
- 657: “But” is changed to “However”.
- 721: “appears” is changed to “appeared”.
- 727: “results in reducing productivity milk” is changed to “reduces cow productivity and”.
- 728: “inducing” is changed to “induces”.
- 733: “is given” is changed to “offered”.

Reviewer 3 Report
Thank you for taking the comments into account when correcting the manuscript.
English language and style are fine/minor spell check is required.
Author Response

(The authors gave the same response as above.)
